# Circulating levels of angiogenic factors and their association with preeclampsia among pregnant women at Mulago National Referral Hospital in Uganda

Sheila Nabweyambo[1☉], Obondo James Sande[2], Naomi McGovern[3], Freddie Bwanga[1], Alfred Ssekagiri[2,4], Annette Keesiga[5], Moses Adroma[6], Ronald Wasswa[7], Maxine Atuheirwe[8], Juliet Namugenyi[8], Barbara Castelnuovo[8], Annettee Nakimuli[6☉]*

1 Department of Medical Microbiology, School of Biomedical Sciences, College of Health Sciences, Makerere University, Kampala, Uganda, 2 Department of Immunology and Molecular Biology, School of Biomedical Sciences, College of Health Sciences, Makerere University, Kampala, Uganda, 3 Department of Pathology, University of Cambridge, Cambridge, United Kingdom, 4 Uganda Virus Research Institute, Entebbe, Uganda, 5 Department of Obstetrics and Gynaecology, Kawempe National Referral Hospital, Kampala, Uganda, 6 Department of Obstetrics and Gynaecology, School of Medicine, College of Health Sciences, Makerere University, Kampala, Uganda, 7 Department of Immunology, Global Health Uganda, Kampala, Uganda, 8 Infectious Diseases Institute, College of Health Sciences, Makerere University, Kampala, Uganda

☉ These authors contributed equally to this work.
* annettee.nakimuli@gmail.com

**Data Availability Statement:** All relevant data are within the paper and its Supporting Information files.

## Abstract

Preeclampsia (PE) is a major cause of maternal and new-born morbidity and mortality. Angiogenic factors contribute a major role in the vascular dysfunction associated with PE. We investigated the circulating levels of vascular endothelial growth factor (VEGF), placental growth factor (PlGF) and soluble Feline McDonough Sarcoma (fms)—like tyrosine kinase-1 (sFlt1), their association with PE and diagnostic performance of disease among pregnant women in Uganda. Using a case-control study design, 106 women with PE and 106 with normal pregnancy were enrolled. Demographic and clinical characteristics, and anticoagulated blood samples were collected from participants. Plasma VEGF, PlGF and sFlt1 levels were measured using Luminex and enzyme linked immunosorbent assays (ELISA). Conditional logistic regression was used to explore association of angiogenic factors with PE and receiver operating characteristic analysis was performed to investigate PE diagnostic performance. Levels of VEGF and PlGF were significantly lower in cases compared to controls (VEGF: median = 0.71 pg/ml (IQR = 0.38–1.11) Vs 1.20 pg/ml (0.64–1.91), p-value<0.001 and PlGF: 2.20 pg/ml (1.08–5.86) Vs 84.62 pg/ml (34.00–154.45), p-value<0.001). Plasma levels of sFlt1 were significantly higher in cases than controls (median = 141.13 (71.76–227.10) x$10^3$ pg/ml Vs 19.86 (14.20–29.37) x$10^3$ pg/ml). Increasing sFlt1 levels were associated with increased likelihood of PE (aOR = 4.73; 95% CI, 1.18–19.01; p-value = 0.0287). The sFlt1/PlGF ratio and sFlt1 had a better performance for diagnosis of PE, with AUC = 0.95 (95% CI, 0.93–0.98) followed by PlGF with AUC = 0.94 (95% CI, 0.91–0.97). Therefore, sFlt1, sFlt1/PlGF ratio

**Funding:** This work was supported by the Makerere University-Uganda Virus Research Institute Centre of Excellence for Infection and Immunity Research and Training (MUII) (https://www.muii.org.ug) through a PhD grant to SN and a group leader award to AN. MUII is supported through the DELTAS Africa Initiative (https://www.aasciences.africa/aesa/programmes/developing-excellence-leadership-training-and-science-africa-deltas-africa) (Grant no. 107743). The DELTAS Africa Initiative is an independent funding scheme of the African Academy of Sciences (AAS), Alliance for Accelerating Excellence in Science in Africa (AESA), and supported by the New Partnership for Africa's Development Planning and Coordinating Agency (NEPAD Agency) with funding from the Wellcome Trust (https://wellcome.ac.uk) (Grant no. 107743) and the UK Government. AN is also supported a NURTURE fellowship (https://www.fic.nih.gov/Grants/Search/Pages/mepi-jr-faculty-TW010132.aspx) (grant number D43TW010132). The funders had no role in study design, data collection and analysis, decision to publish, or preparation of the manuscript.

**Competing interests:** The authors have declared that no competing interests exist.

and PIGF are potential candidates for incorporation into algorithms for PE diagnosis in the Ugandan population.

## Introduction

Preeclampsia (PE) is a pregnancy complication that affects 3–8% pregnant women worldwide, with a higher burden among African women [1, 2]. Occurring at 20 and above weeks of gestation, PE presents with new onset high blood pressure and proteinuria in previously normotensive women [3]. Preeclampsia is associated with adverse complications including; HELLP syndrome (hemolysis, elevated liver enzymes, low platelet count), eclampsia and mortality for the mother, as well as preterm birth, low birth weight, morbidity and mortality for the baby [1, 3–5].

The pathophysiology of PE is suggested to be triggered by release of placental factors into maternal circulation, that cause systemic vascular dysfunction. The systemic vascular dysfunction is associated with impaired angiogenesis, vascular permeability and endothelial cell function, which culminate into high blood pressure, exaggerated systemic inflammatory response and end organ damage (such as the kidney and liver) [6, 7]. The preeclamptic placenta releases excess amounts of antiangiogenic soluble Feline McDonough Sarcoma (fms)- like tyrosine kinase-1 (sFlt-1) into maternal circulation, which binds free circulating proangiogenic Vascular endothelial growth factor (VEGF) and Placental growth factor (PlGF), leading to an antiangiogenic state [8, 9]. VEGF and PlGF promote vascular development by activating endothelial cell proliferation and migration, and maintenance of vascular integrity. Binding by sFlt1 prevents VEGF and PlGF signalling via the VEGF-1 receptor on endothelial cells, thereby disrupting endothelial cell activation, leading to abnormal function and hence vascular dysfunction [10, 11].

Increasing research indicates that women with PE present with elevated circulating sFlt1 levels and significantly reduced concentrations of free VEGF and PlGF compared to normal pregnant women. This profile occurs before onset of clinical disease, in addition to these angiogenic factors being differentially expressed in the various PE phenotypes and other pregnancy complications, such as gestational hypertension [12, 13]. Angiogenic factors (sFlt1/PlGF ratio) were also found to out-perform the standard PE diagnosis procedure (including; blood pressure (BP) measurement, detection of protein in urine and varying array of clinical measurements and laboratory biochemical analyses) [14]. Hence angiogenic factors have increasingly become a target for potential use as markers of prediction, confirmation of PE diagnosis and disease progression [10, 15–18].

In a study by Rana, S et al. [19], symptomatic women with a normal angiogenic profile recorded no adverse pregnancy outcomes, suggesting the usefulness of angiogenic factors to identify women less likely to develop PE related complications. This would enable appropriate allocation of interventions to high risk women. In the wake of limited resources, Uganda- a sub Saharan African country, faces a high burden of PE and could benefit from the use of angiogenic factors for better management of PE patients [20]. However, there is limited documentation on the circulating levels of angiogenic factors associated with PE among pregnant women in Uganda. The aim of this study was to investigate the circulating levels of angiogenic factors (sFlt1, VEGF, and PlGF) and their association to PE in a Ugandan population, and to determine their performance for diagnosis of PE among pregnant women.

## Materials and methods

### Study design and study population

This was a case control study conducted from March to October 2019 at Mulago national referral hospital in Kampala, Uganda. Cases were women with PE and controls were normotensive pregnant women. Preeclampsia was defined as new onset hypertension consisting of systolic BP $\geq$ 140 mmHg and diastolic BP $\geq$ 90 mmHg on 2 different measurements at least 4 hours apart plus proteinuria $\geq$ +1 on urine dipstick, at 20 and above gestation weeks [21]. Controls were defined as normal pregnant women presenting with systolic BP < 140 mmHg and diastolic BP< 90 mmHg on the first, or second measurement after 4 hours, with trace or negative protein in urine. Cases were recruited form the maternity ward and controls from the outpatient antenatal clinics. Controls were matched to cases by maternal and gestational age, at a ratio of 1:1 using the following categories; 18–22, 23–27, 28–32, and above 32 years, and 20–28, 29–33, 34–37, 38–42 weeks of gestation, so as to obtain better interpretation of the results [22].

This study excluded women who had confirmed fetal abnormalities and pre-existing pathologies (including; diabetes mellitus, chronic hypertension, cardio vascular disease, chronic renal disease).

### Sample size estimation

Prior data indicated that black African women with PE had significantly lower plasma levels of PlGF (90.3±8.9pg/ml versus 172.8±20.2pg/ml; p<0.01), higher sFlt1 (2087.3±200.1pg/ml versus 1546.5 ± 91.9 pg/ml; p<0.01) and a higher sFlt1/PlGF ratio (66.8 ± 18.7 versus 22.3 ± 2.9; p<0.01) compared to normotensive controls [16]. Using these data, and assuming 80% study power and 0.05 type I error probability associated with the test of the null hypothesis; for a one control per case; an estimated number of cases equal to 74; and hence 74 controls was obtained. This was computed using the formula for calculating sample size to detect a significant difference between two means with equal sample sizes and unequal variances as implemented by Epitools [23]. However, 106 case/control pairs were enrolled to allow for failure to obtain required samples and loss to follow up.

### Study procedures

A study data tool was used to obtain demographic data, and clinical characteristics of the study participants. Data collected included: maternal and gestational age, marital status, religion, education level, type of pregnancy, alcohol consumption, smoking history, parity, HIV status, family history of PE and hypertension, and diagnosis with PE in a previous pregnancy. From each study participant, 6mL of ethylenediaminetetraacetate (EDTA) anti-coagulated blood sample were collected by a trained study midwife or nurse. Blood samples were transported within one hour to the Translational Research laboratory at the Infectious Diseases Institute, Makerere university, Kampala, Uganda.

This study was approved by the School of Biomedical Sciences Higher Degrees Research and Ethics Committee (REF: SBS-HDREC-617) and the Uganda National Council for Science and Technology (REF: HS 2535), and all study participants provided written informed consent before enrolment into the study.

### Plasma collection and immunoassays

Upon laboratory reception, blood samples were centrifuged at 6˚C at a speed of 1008xg for 15 minutes. The supernatants were collected in sterile falcon tubes and centrifuged again at

1500xg for 10 minutes at 6°C. Resulting platelet poor plasma per sample was separated into two vials and stored at -80°C for later use to perform the immunoassays (Fig 1).

Stored plasma samples were thawed at 4–8°C. Immunoassays were performed in duplicates using VEGF and PlGF Magnetic luminex performance assay (Human Angiogenesis Premixed KitA; R&D Systems, a bio-techne brand), and Human VEGF R1/Flt-1 Quantikine ELISA Kit (R&D Systems), following manufacturer's instructions. Plasma levels of angiogenic factors were measured in pg/ml, and validated commercial control samples for each analyte were assayed in parallel as a quality control measure. Commercial control samples used include; Luminex Performance Assay, Human Angiogenesis Panel A Kit Controls (R&D Systems, a bio-techne brand) for VEGF and PlGF luminex assays, and Quantikine Immunoassay Control Set 935 for Human VEGF R1 (R&D Systems, a bio-techne brand) for sFlt1 ELISA assays. Inter-assay coefficients of variation for VEGF and PlGF was 10.03% and 9.72% respectively whereas that of sFlt1 was 8.08%.

## Statistical analysis

Categorical variables (demographic and clinical characteristics) were summarised as absolute numbers and proportions. Continuous variables (plasma concentration of angiogenic factors) were summarised as median and interquartile ranges. Mann-Whitney U-test was used to compare continuous variables and Fisher's exact test for comparing categorical variables. The association of angiogenic factors with PE was explored using conditional logistic regression. The diagnostic performance of angiogenic factors for PE was investigated by training logistic regression models at 10-fold cross validation. Performance was evaluated using five metrics; specificity, sensitivity, positive predictive value, negative predictive value and area under the receiver operating characteristic curve. All analyses were performed in R statistical programming environment version 4.0.2 [24]. Packages used for analysis include; survival [25] for conditional logistic regression, caret [26] for building predictive models, pROC [27] for ROC analysis and ggplot2 [28] for visualization of results.

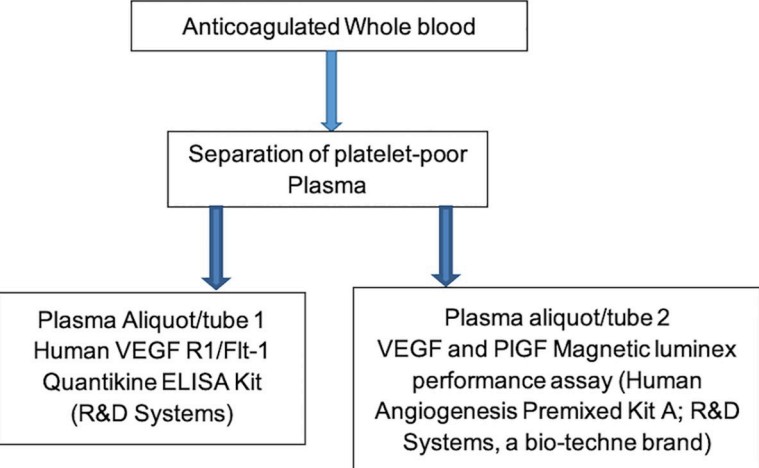

**Fig 1. Chart showing flow of laboratory procedures.** Platelet poor plasma was processed from ethylenediaminetetraacetate anticoagulated whole blood and divided into 2 aliquots that were used during the study immunoassays (Luminex and ELISA). ELISA = enzyme linked immunosorbent assay.

## Results

### Study participant characteristics

Demographic and clinical characteristics of study participants are shown in Tables 1 and 2 respectively. The mean age of the study population was 25.6 and 25.7 years among the cases and controls respectively. Majority of the study participants were married and had attained secondary level education. All participants self-reported to be Blacks Africans with both parents of African ancestry. Further ancestral delineation was not feasible. All participants had conceived naturally.

### Plasma levels of sFlt1, PIGF and VEGF

The distribution and Plasma levels of angiogenic factors between cases and controls are shown in Fig 2 and Table 3 respectively. Plasma levels of VEGF and PlGF were significantly lower in cases compared to controls (median = 0.71 pg/ml (IQR = 0.38–1.11) Vs 1.20 pg/ml (0.64–1.91) and 2.20 pg/ml (1.08–5.86) Vs 84.62 pg/ml (34.00–154.45) respectively), while there were significantly higher levels of sFlt1 in cases than in controls (141.13 (71.76–227.10) x10$^3$ pg/ml Vs

**Table 1. Demographic characteristics of study participants.**

| Demographic characteristics N (%) | | | |
|---|---|---|---|
| Characteristic | Controls (N = 106) | Cases (N = 106) | p-value |
| **Age group** | | | 0.8110 |
| 18–22 | 35 (33.0) | 34 (32.1) | |
| 23–27 | 28 (26.4) | 32 (30.2) | |
| 28–32 | 30 (28.3) | 30 (28.3) | |
| Above 32 | 13 (12.3) | 10 (9.4) | |
| **Marital status** | | | 0.5001 |
| Married | 97 (91.5) | 93 (87.7) | |
| Not Married | 9 (8.5) | 13 (12.3) | |
| **Religion** | | | 0.8766 |
| Catholic | 28 (26.4) | 24 (22.6) | |
| Protestant | 29 (27.4) | 26 (24.5) | |
| Muslim | 25 (23.6) | 29 (27.3) | |
| Seventh day Adventist | 3 (2.8) | 4 (3.8) | |
| Others | 21 (19.8) | 23 (21.7) | |
| **Education level** | | | **0.0105** |
| None | 1 (1.0) | 0 (0.0) | |
| P(1–4) | 5 (4.7) | 6 (5.7) | |
| P(5–7) | 21 (19.8) | 31 (29.2) | |
| S(1–4) | 59 (55.7) | 34 (32.1) | |
| S(5–6) | 8 (7.5) | 12 (11.3) | |
| Tertiary/university | 12 (11.3) | 23 (21.7) | |
| **Gestational age categories (weeks)** | | | 1.0000 |
| 20–28 | 9 (8.5) | 9 (8.5) | |
| 29–33 | 29 (27.4) | 29 (27.4) | |
| 34–37 | 27 (25.5) | 27 (25.5) | |
| 38–42 | 41 (38.7) | 41 (38.7) | |

P, Primary education; S, Secondary education; N, Number.

**Table 2. Clinical characteristics of study participants.**

| Clinical characteristics n (%) | | | |
|---|---|---|---|
| Characteristic | Controls (N = 106) | Cases (N = 106) | p-value |
| **Type of pregnancy** | | | 0.1188 |
| Singleton | 104 (98.1) | 100 (94.3) | |
| Multiple | 1 (1.0) | 6 (5.7) | |
| Unknown | 1 (1.0) | 0 (0.0) | |
| **Medical History** | | | |
| **Alcohol consumption** | | | 0.5936 |
| No | 100 (94.3) | 98 (92.5) | |
| Yes | 6 (5.7) | 8 (7.5) | |
| **HIV status** | | | 0.1345 |
| Negative | 97 (91.5) | 103 (97.2) | |
| Positive | 9 (8.5) | 3 (2.8) | |
| **Family history of PE** | | | 0.1703 |
| No | 104 (98.11) | 99 (93.4) | |
| Yes | 2 (1.9) | 7 (6.6) | |
| **Family history of Hypertension** | | | **0.0534** |
| No | 80 (75.5) | 66 (62.3) | |
| Yes | 26 (24.5) | 40 (37.7) | |
| **Family history of diabetes melitus** | | | 0.4593 |
| No | 86 (81.1) | 91 (85.8) | |
| Yes | 20 (18.9) | 15 (14.2) | |
| **Obstetric history** | | | |
| **Parity** | | | 0.3965 |
| 1 | 69 (65.1) | 62 (58.5) | |
| More than 1 | 37 (34.9) | 44 (41.5) | |
| **Diagnosis with Hypertension in previous pregnancy** | | | **0.0128** |
| No | 65 (61.3) | 48 (45.3) | |
| Yes | 4 (3.8) | 14 (13.2) | |
| Not applicable | 37 (34.9) | 44 (41.5) | |
| **Smoking history** | | | 1.0000 |
| No | 106 (100.0) | 105 (99.0) | |
| Yes | 0 (0.0) | 1 (1.0) | |

PE, Preeclampsia; N, Number.

19.86 (14.20–29.37) x$10^3$ pg/ml). The sFlt1/PlGF ratio was significantly higher in the cases compared to controls (p-value< 0.001).

## Plasma levels of sFlt1, PlGF and VEGF across gestational age categories

The distribution of plasma levels of VEGF, PlGF and sFlt1 across gestational age (GA) categories (in weeks) is shown in Fig 3 and Table 4. The trend of plasma VEGF levels consists of 2 high peaks at weeks 20–28 and 34–37 in both cases and controls. In both cases and control groups, VEGF plasma levels drop from 20–28 to 29–33 weeks. The second peak declines from 34–37 to 38–42 weeks in both cases and controls respectively. Significant difference of plasma VEGF levels between cases and controls was only observed for GA categories 29–33 weeks (p-value = 0.0033), and 34–37 weeks (p-value = 0.0017).

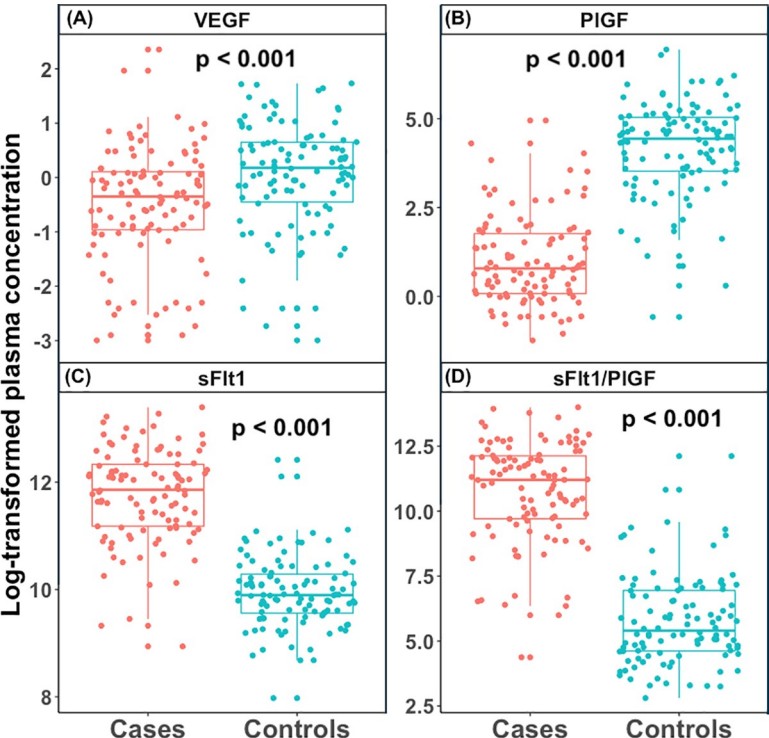

**Fig 2. Plasma levels of angiogenic factors.** (**A**) Logarithm of the median plasma levels of vascular endothelial growth factor (VEGF) in cases Vs controls, (**B**) logarithm of the median plasma levels of placental growth factor (PlGF) in cases Vs controls, (**C**) logarithm of the median plasma levels of soluble fms-like tyrosine kinase-1 (sFlt-1) in cases Vs controls, (**D**) logarithm of the median plasma levels of and sFlt-1/PlGF ratio in cases Vs controls. The Boxes represent interquartile range, with the top and bottom horizontal lines indicating the 75th and 25th percentiles of the data for each group (cases and controls). The horizontal line within the box represents the median value. Significant differences between cases and controls was observed for each angiogenic factor (p-value <0.001).

Plasma PlGF levels were significantly lower in cases compared to controls, across all GA categories (p-value<0.001). In the cases, PlGF plasma levels increase from 20–28 to 29–33 weeks, from which the trend declines at 34–37 weeks, and increases again at 38–42 weeks. In the controls, the trend consists of 2 high peaks at 20–28 and 34–37 weeks, which decline at 29–33 and 38–42 weeks respectively.

Plasma levels of sFlt1 were significantly higher in cases across all GA categories as compared to the controls (p-value<0.001). Plasma sFlt1 levels gradually increase in the controls from 20–28 to 34–37 weeks and slightly declines at 38–42 weeks. The cases indicate 2 high

**Table 3. Plasma levels of VEGF, PlGF and sFlt1 between cases and controls.**

| Angiogenic factors | Controls Median (IQR) | Cases Median (IQR) | p-value |
|---|---|---|---|
| VEGF (pg/ml) | 1.20 (0.64–1.91) | 0.71 (0.38–1.11) | < 0.001 |
| PlGF (pg/ml) | 84.62 (34.00–154.45) | 2.20 (1.08–5.86) | < 0.001 |
| sFlt1 x $10^3$ (pg/ml) | 19.86 (14.20–29.37) | 141.33 (71.76–227.10) | < 0.001 |
| sFlt1/PlGF x $10^3$ | 0.222 (0.101–0.104) | 73.37 (16.46–184.80) | < 0.001 |

PlGF, placental growth factor; VEGF, vascular endothelial growth factor; sFlt1, soluble fms like tyrosine kinase; sFlt1/PlGF, soluble fms like tyrosine kinase/placental growth factor ratio.

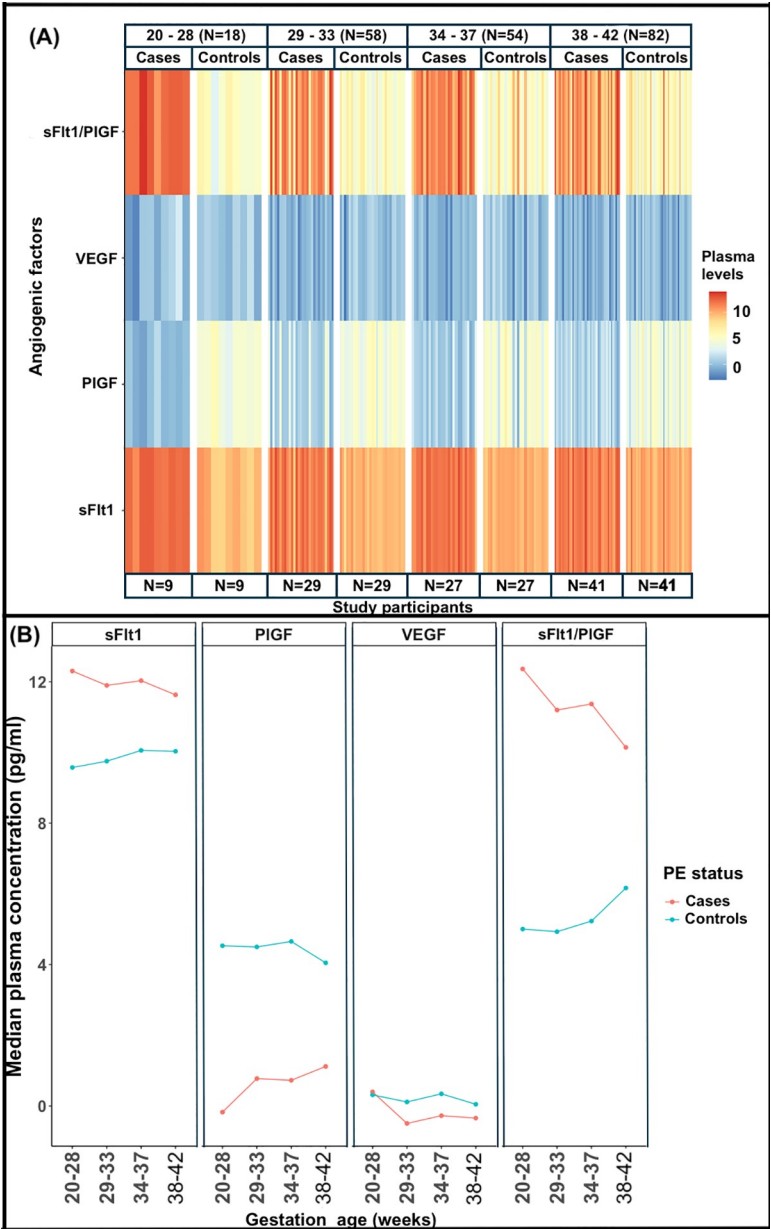

**Fig 3. Distribution of angiogenic factors across GA categories.** (**A**) Heat map showing the distribution of logarithm plasma levels of angiogenic factors between cases and controls groups across the GA categories, (**B**) trend curves of log transformed plasma concentrations of each angiogenic factor across GA categories, between cases and controls. sFlt1, PlGF and sFlt1/PlGF ratio were significantly different at all GA categories.

peaks of plasma sFlt1 levels at 20–28 and 34–37 weeks from which they drop at 29–33 and 34–37 weeks respectively.

## sFlt1/PlGF ratio across gestational age groups

The sFlt1/PlGF ratio was significantly higher in cases compared to controls across all gestational age categories (p-value< 0.001) (Fig 3, Table 4). The sFlt1/PlGF ratio is highest at 20–28 weeks for pre-eclamptic women and at 38–42 weeks for normotensive pregnant controls.

**Table 4. Plasma levels of angiogenic factors between cases and controls by gestational age groups.**

| Angiogenic factors | GA categories (weeks) | Controls (IQR) | Cases (IQR) | p-value |
|---|---|---|---|---|
| VEGF (pg/ml) | 20–28 | 1.37 (0.94–2.81) | 1.49 (0.39–1.62) | 0.6496 |
| | 29–33 | 1.12 (0.78–2.17) | 0.61 (0.29–1.12) | **0.0033** |
| | 34–37 | 1.41 (0.81–2.62) | 0.76 (0.45–1.06) | **0.0017** |
| | 38–42 | 1.05 (0.41–1.54) | 0.71 (0.36–1.09) | 0.2784 |
| PlGF (pg/ml) | 20–28 | 92.89 (65.42–104.95) | 0.84 (0.60–0.99) | **< 0.001** |
| | 29–33 | 89.87 (36.43–169.16) | 2.17 (1.31–17.52) | **< 0.001** |
| | 34–37 | 104.95 (42.68–158.10) | 2.07 (1.03–4.16) | **< 0.001** |
| | 38–42 | 57.36 (22.37–131.28) | 3.06 (1.51–6.52) | **< 0.001** |
| sFlt1 x10³ (pg/ml) | 20–28 | 14.39 (6.45–25.88) | 220.54 (171.75–259.30) | **< 0.001** |
| | 29–33 | 17.21 (12.90–20.42) | 146.90 (69.02–198.25) | **< 0.001** |
| | 34–37 | 23.34 (14.58–29.01) | 167.95 (92.58–216.02) | **< 0.001** |
| | 38–42 | 22.77 (17.74–37.83) | 112.39 (66.35–191.51) | **< 0.001** |
| sFlt1/ PlGF ratio (x10³) | 20–28 | 0.15 (0.10–0.36) | 233.81 (155.64–314.20) | **< 0.001** |
| | 29–33 | 0.14 (0.09–0.37) | 73.23 (7.02–173.36) | **< 0.001** |
| | 34–37 | 0.19 (0.10–0.44) | 86.85 (30.27–151.18) | **< 0.001** |
| | 38–42 | 0.48 (0.16–1.27) | 25.41(10.09–158.62) | **< 0.001** |

All results are presented as median (IQR (Interquartile range))

GA, gestational age; PlGF, placental growth factor; VEGF, vascular endothelial growth factor; sFlt1, soluble fms like tyrosine kinase; sFlt1/PlGF, soluble fms like tyrosine kinase/placental growth factor ratio.

## Association of angiogenic factors with preeclampsia

To determine whether plasma levels of VEGF, PlGF and sFlt1 are associated with PE, we performed univariate and multivariate conditional logistic regression. Prior to analysis, plasma concentration of all angiogenic factors was log transformed to the base of two, as such the odds ratios correspond to effects of doubling the concentration of plasma for respective angiogenic factors. Variables with p-value less than 0.25 in univariate analysis were considered in the multivariate analysis and the variables used in the final model were obtained by backward elimination. In the multivariate analysis, we adjusted for family history of PE and family history of hypertension. Increase in plasma concentration of sFlt1 was significantly associated with increased likelihood of PE (aOR = 4.73, 95% CI; 1.18–19.01, p-value = 0.0287) and increase in plasma concentration of PlGF was significantly associated with decreased likelihood of PE (aOR = 0.40, 95% CI; 0.20–0.81, p-value = 0.0101) (Table 5).

**Table 5. Angiogenic factors associated with preeclampsia.**

| Angiogenic factors | Crude OR (95% CI) | p-value | Adjusted OR (95% CI) | p-value |
|---|---|---|---|---|
| VEGF | 0.67 (0.54–0.84) | < 0.001 | 1.64 (0.74–3.66) | 0.2259 |
| PlGF | 0.39 (0.25–0.59) | < 0.001 | 0.40 (0.20–0.81) | **0.0104** |
| sFlt1 | 5.47 (2.59–11.55) | < 0.001 | 4.73 (1.18–19.01) | **0.0287** |
| sFlt1/ PlGF ratio | 2.18 (1.48–3.22) | < 0.001 | – | – |
| Family history of hypertension* | 2.00 (1.05–3.80) | < 0.001 | 0.08 (0.00–3.10) | 0.1730 |
| Family history of PE* | 3.50 (0.73–16.85) | 0.1180 | 12.73 (0.26–630.13) | 0.2014 |

PlGF, placental growth factor; VEGF, vascular endothelial growth factor; sFlt1, soluble fms like tyrosine kinase; sFlt1/PlGF, soluble fms like tyrosine kinase/placental growth factor ratio

*Variables adjusted for in the multivariate analysis.

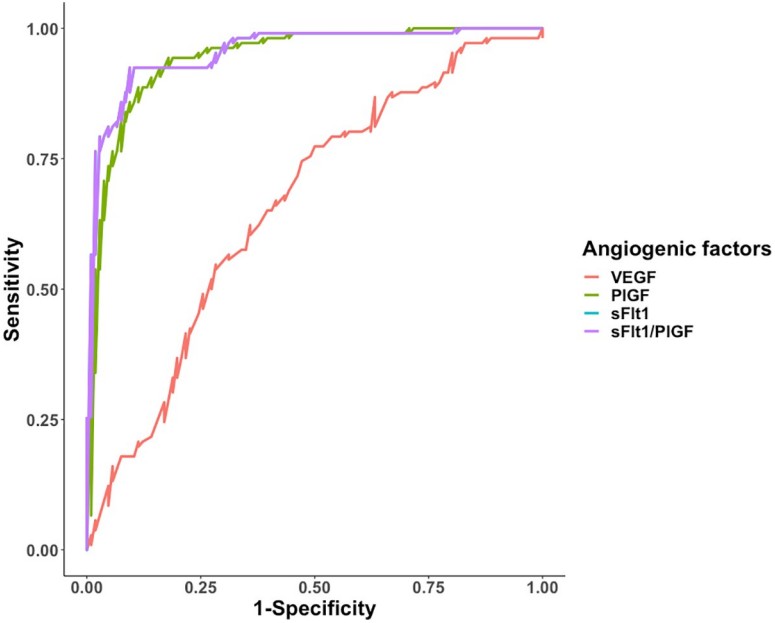

**Fig 4. Predictive characteristics of angiogenic factors.** ROC analysis of predictive behaviour of VEGF, PlGF, sFlt1, and sFlt1/PlGF ratio for PE, Area under curve (AUC) for sFlt1 and sFlt1/PlGF ratio is 0.95, followed by PlGF at 0.94 and lastly VEGF at 0.66.

### Diagnostic performance of angiogenic factors

We set out to investigate the ability of VEGF, sFlt1, PlGF and sFlt1/PlGF ratio to diagnose PE by assessing related sensitivity and specificity, plus the negative and positive predictive values. Logistic regression models were trained and evaluated at 10-fold cross validation. The sFlt1/PlGF ratio had the highest sensitivity (0.92, 95% CI; 0.87–0.97) followed by sFlt1 (0.89, 95% CI; 0.83–0.95) and PlGF (0.86, 95% CI 0.79–0.92). VEGF had the least sensitivity (0.56, 95% CI; 0.46–0.65). In the same light, sFlt1/PlGF ratio had the highest specificity (0.91, 95% CI; 0.85–0.96) followed by PlGF (0.90, 95% CI; 0.84–0.95) and sFlt1 (0.89, 95% CI 0.83–0.95). VEGF had the least specificity (0.69, 95% CI; 0.60–0.77). Results of ROC analysis show that sFlt1 and the sFlt1/PlGF ratio had the highest AUC (0.95, 95% CI; 0.93–0.98) followed by PlGF (0.94, 95% CI; 0.91–0.97). VEGF had the least AUC (0.66, 95% CI; 0.59–0.73) (Fig 4, Table 6).

## Discussion

The pathogenesis of PE is not fully understood given its heterogeneous clinical presentation and laboratory findings. This makes prevention, prediction, early detection and management

**Table 6. Predictive characteristics of angiogenic factors.**

| Angiogenic factors | Sensitivity (95% CI) | Specificity (95% CI) | PPV (95% CI) | NPV (95% CI) | AUC (95% CI) | Probability cut-off values |
|---|---|---|---|---|---|---|
| VEGF | 0.56 (0.46–0.65) | 0.69 (0.60–0.77) | 0.64 (0.54–0.74) | 0.61 (0.52–0.70) | 0.66 (0.59–0.73) | 0.4592 |
| PlGF | 0.86 (0.79–0.92) | 0.90 (0.84–0.95) | 0.89 (0.83–0.95) | 0.86 (0.80–0.93) | 0.94 (0.91–0.97) | 0.4534 |
| sFlt1 | 0.89 (0.83–0.95) | 0.89 (0.83–0.95) | 0.89 (0.83–0.95) | 0.89 (0.83–0.95) | 0.95 (0.93–0.98) | 0.4359 |
| sFlt1/ PlGF ratio | 0.92 (0.87–0.97) | 0.91 (0.85–0.96) | 0.91 (0.85–0.96) | 0.92 (0.87–0.97) | 0.95 (0.93–0.98) | 0.4359 |

PlGF, placental growth factor; VEGF, vascular endothelial growth factor; sFlt1, soluble fms like tyrosine kinase; sFlt1/PlGF, soluble fms like tyrosine kinase/placental growth factor ratio; PPV, positive predictive value; NPV, negative predictive value; AUC, Area under curve; CI, confidence interval.

a challenge. Angiogenic factors constitute a list of biomarkers suggested for their potential use in the management of PE in different populations [16, 29–31]. In this study, we set out to determine the circulating levels of angiogenic factors (sFlt1, VEGF, and PlGF) and their association with PE among women in a Ugandan population, and to determine their ability to diagnose PE.

We obtained significant differences in the plasma levels of VEGF, PlGF and sFlt1, and the sFlt1/PlGF ratio between cases and controls. Compared to controls, cases had reduced plasma levels of VEGF and PlGF, increased levels of sFlt1, and higher sFlt1/PlGF ratio. These results show an aberrant balance of the angiogenic factors among the women with PE compared to the normal pregnant women, which is in agreement with findings from earlier studies [13, 15, 16]. A study carried out in the United states among nulliparous women found lower VEGF and PlGF levels, and higher sFlt1 levels in serum samples obtained from women with symptoms of clinical PE, as compared to those without PE. Reduced free VEGF and PlGF levels was attributed to binding by the increased sFlt-1 levels [13]. This is similar to findings from a study carried out among South African black women, in which serum PlGF and sFlt1 levels were respectively lower and higher among the women with PE than in healthy pregnant women [16]. Reduced serum VEGF levels were also found among Tunisian women with PE compared to healthy pregnant controls, in addition to reduced PlGF and higher sFlt1 levels [15].

Contrary to our results, some studies observed increased VEGF serum levels among women presenting with PE [29, 32]. This may be attributed to inherent population differences, different methodological approaches, and the fact that platelets release VEGF during the clotting process, hence increasing its levels in serum samples. Similarly, varying methodological approaches such as laboratory assays, and inherent population differences may explain the contrast in the angiogenic factor levels obtained in our study, compared to results from earlier studies [12, 13, 15, 16, 33]. The validity of our study results was ascertained through the use of validated commercial control samples in all laboratory assays performed.

We further determined the plasma levels of the angiogenic factors across defined gestational age ranges. In the controls, the plasma levels of both PlGF and VEGF were found to be higher at until 37 weeks and then declined from 38 weeks onwards. Given the role of both VEGF and PlGF in vascular development, decline in their circulating levels after 38 weeks could be explained by the already established maternal-placental circulation around the mid gestation [34]. The sFlt1 levels are shown to gradually increase across all GA ranges towards term in the controls group; this occurs in normal pregnancy due to increasing placental ischemia and oxidative stress as the pregnancy progresses [35, 36]. Increased sFlt1 levels may partly explain the decreasing levels of both VEGF and PlGF towards the end of pregnancy due to the anti-angiogenic role of sFlt1 [15, 34]. We further observed that PlGF levels were lowest, and sFlt1 levels highest in the cases at 20–28 weeks of gestation, compared to those with PE at 29 weeks and above. This may correspond to early onset PE which has been associated with more pronounced alterations in the circulating angiogenic factors; including much lower PlGF and higher sFlt1 levels compared to late onset PE [37, 38]. This trend is also associated with adverse complications observed in early onset PE [12]. Significantly lower VEGF levels between cases and controls was only observed at gestational weeks 29–37, which is similar to results obtained by Gannoun, M et al. [15].

At multivariate analysis, increase in plasma levels of sFlt1 was significantly associated with increased risk of developing PE while increase in PlGF plasma levels was significantly associated with decreased risk of PE. This is similar to what has been found by other studies [13, 15, 30, 39]. Placental release of excess amounts of sFlt1 in maternal circulation results into wide spread vascular endothelial dysfunction [40]. Binding of PlGF by the excess sFlt1 levels disrupts placental vascular development, leading to placental ischemia observed in PE [15, 41].

Due to the often very low plasma levels of free VEGF found throughout pregnancy in most studies, PlGF which is a member of the VEGF family, sFlt1 and the sFlt1/PlGF ratio have emerged as potential biomarkers for use in prediction, diagnosis and prognosis of PE [11, 12, 16, 42]. Using ROC analysis in this study, the sFlt1/PlGF ratio had better performance for diagnosis of PE, followed by sFlt1 and PlGF. This is similar to results obtained in other previous studies [42–44]. A study carried out among women with singleton pregnancies in china obtained an AUROC of 0.98 (95% CI, 0.969–1.000) for the PE diagnostic performance of the sFlt-1/PlGF ratio [42]. Similarly, the diagnosis of PE using a novel approach that employed the sFlt1/PlGF ratio accurately reduced the number of patients identified as pre-eclamptic using the standard method (BP, proteinuria and laboratory tests; alanine aminotransferase, and platelet counts) from 42% to 4%. This suggests the possible use of plasma sFlt1/PlGF ratio for better risk stratification of patients with suspected PE. In the same study, use of sFlt1/PlGF ratio reduced the cost of managing a PE patient to $540-$1215 from $3022 at Beth Israel Deaconess Medical Center in Boston, Massachusetts [44]. A pilot study among pregnant women in Mozambique demonstrated that low PlGF levels in early pregnancy were associated with adverse maternal and perinatal outcomes, and increased transfers to higher level care [18]. Therefore, introducing angiogenic factors sFlt1 and PlGF into routine clinical care of women in Uganda may; 1) improve risk stratification and diagnosis compared to the standard diagnostic criteria of PE, 2) guide decision making regarding close surveillance of those at high risk of adverse outcomes, especially those with pronounced alterations of plasma PlGF and sFlt1.

One of the limitations in our study is that we were unable to analyse the diagnostic performance of the angiogenic factors at the different GA categories due to the limited sample size. The cross-sectional study design used could only allow for collection of end point samples, hence we were unable to evaluate the predictive power of angiogenic factors for risk of PE.

## Conclusions and recommendations

Our study supports the hypothesis that an imbalance of circulating angiogenic factors contributes to the pathophysiology of PE. We further demonstrate that the sFlt1/PlGF ratio, sFlt1 and PlGF have a high diagnostic ability for PE, and hence potential candidates for incorporation into a diagnostic algorithm for PE.

We recommend prospective longitudinal studies to enable measurement of the circulating angiogenic factors at different gestational time points throughout pregnancy. This will enable better evaluation of the predictive and diagnostic ability of the angiogenic factors for PE, as well as their potential use for management and prevention of adverse outcomes of PE among pregnant women in Uganda.

## Supporting information

**S1 Dataset. Study data.**
(CSV)

**S1 Datatool. Study data tool.**
(DOC)

## Acknowledgments

We thank the staff of the Translational Research Laboratory at the Infectious Diseases Institute for their support towards performing the laboratory procedures and assays. We acknowledge the support given by the study team; Rosemary Byenkya, Margaret Sewagaba, Noela Kalyowa, Emily Nakirijja, Moreen Ahimbisibwe, Elizabeth Mutesi, Rita Kukundakwe, Florence

Walabyeki, Ruth Namubiru and Doreen Birungi. Lastly we acknowledge the expertise of Frank Mubiru at the Infectious Diseases Institute, who participated in the development of the study database and analysis of study data.

## Author Contributions

**Conceptualization:** Sheila Nabweyambo, Obondo James Sande, Naomi McGovern, Freddie Bwanga, Annettee Nakimuli.

**Data curation:** Sheila Nabweyambo, Obondo James Sande, Naomi McGovern, Alfred Ssekagiri, Annette Keesiga, Moses Adroma, Maxine Atuheirwe, Annettee Nakimuli.

**Formal analysis:** Sheila Nabweyambo, Obondo James Sande, Alfred Ssekagiri, Annettee Nakimuli.

**Funding acquisition:** Sheila Nabweyambo, Annettee Nakimuli.

**Investigation:** Sheila Nabweyambo, Obondo James Sande, Naomi McGovern, Annette Keesiga, Moses Adroma, Ronald Wasswa, Maxine Atuheirwe, Juliet Namugenyi, Annettee Nakimuli.

**Methodology:** Sheila Nabweyambo, Obondo James Sande, Naomi McGovern, Freddie Bwanga, Annettee Nakimuli.

**Project administration:** Sheila Nabweyambo, Annette Keesiga, Moses Adroma.

**Resources:** Annettee Nakimuli.

**Software:** Alfred Ssekagiri.

**Supervision:** Sheila Nabweyambo, Obondo James Sande, Naomi McGovern, Freddie Bwanga, Annette Keesiga, Moses Adroma, Annettee Nakimuli.

**Validation:** Sheila Nabweyambo, Obondo James Sande, Naomi McGovern, Freddie Bwanga, Alfred Ssekagiri, Ronald Wasswa, Juliet Namugenyi, Barbara Castelnuovo, Annettee Nakimuli.

**Visualization:** Sheila Nabweyambo, Obondo James Sande, Naomi McGovern, Freddie Bwanga, Alfred Ssekagiri, Barbara Castelnuovo, Annettee Nakimuli.

**Writing – original draft:** Sheila Nabweyambo, Obondo James Sande, Alfred Ssekagiri, Annettee Nakimuli.

**Writing – review & editing:** Sheila Nabweyambo, Obondo James Sande, Naomi McGovern, Freddie Bwanga, Alfred Ssekagiri, Annette Keesiga, Moses Adroma, Ronald Wasswa, Maxine Atuheirwe, Juliet Namugenyi, Barbara Castelnuovo, Annettee Nakimuli.

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
