## [Decision Letter · Decision Letter 0]

29 Dec 2020

PONE-D-20-27867

Circulating levels of angiogenic factors and their association with preeclampsia among pregnant women at Mulago National Referral Hospital in Uganda

PLOS ONE

Dear Dr. Nakimuli,

Thank you for submitting your manuscript to PLOS ONE. After careful consideration, we feel that it has merit but does not fully meet PLOS ONE’s publication criteria as it currently stands. Therefore, we invite you to submit a revised version of the manuscript that addresses the points raised during the review process.

We look forward to receiving your revised manuscript.

Kind regards,

Antonio Simone Laganà, M.D., Ph.D.

Academic Editor

PLOS ONE

Additional Editor Comments:

The topic of the manuscript is interesting. Nevertheless, the reviewers raised several concerns: considering this point, I invite authors to perform the required major revisions.

Journal Requirements:

3. You indicated that you had ethical approval for your study. In your Methods section, please ensure you have also stated whether you obtained consent from parents or guardians of the minors included in the study or whether the research ethics committee or IRB specifically waived the need for their consent.

4. Please provide a sample size and power calculation in the Methods, or discuss the reasons for not performing one before study initiation.

Reviewers' comments:

Reviewer's Responses to Questions

**Comments to the Author**

1. Is the manuscript technically sound, and do the data support the conclusions?

Reviewer #1: Yes

Reviewer #2: Yes

Reviewer #3: Partly

2. Has the statistical analysis been performed appropriately and rigorously? 

Reviewer #1: Yes

Reviewer #2: Yes

Reviewer #3: No

3. Have the authors made all data underlying the findings in their manuscript fully available?

Reviewer #1: Yes

Reviewer #2: Yes

Reviewer #3: Yes

4. Is the manuscript presented in an intelligible fashion and written in standard English?

Reviewer #1: Yes

Reviewer #2: Yes

Reviewer #3: Yes

5. Review Comments to the Author

Reviewer #1: THE STUDY HAS BEEN CONDUCTED WELL TO ANALYSE THE FACT THAT THESE ANGIO MARKERS ARE RAISED IN PE . THe predictability is not assessed due to the fact that the inclusion criteria has been a 20 weeks of gestation . so the narration about this issue is out of context and needs to be modified . establishment of the fact that angiogenic markers are deranged in comparison to matched normal population is a good study . What you can do is look at the levels quantitatively and identify those one who have a count of 85 or more and compare them to find out whether they hada severe disease . this way rather than for prediction you can also understand its value in risk stratification whihc is clinically of great value . If you are interested you can request the gynecs in your institution to connect to me at girijawagh@gmail.com about a gestosis score validation may be can be done in-the same population through a different study with new ethics approval .

Reviewer #2: This is a case control study that reflects the association of maternal circulating levels of VEGF, PlGF, sFlt1 and the sFlt1/PlGF ratio with PE, and compare their distribution both in normal healthy pregnancies as well as those with PE and to determine their predictive power of the disease among women in a Ugandan population.

,

The uniqueness of the study is employing different methodological approaches such as Luminex assay ,heat map, trained logistical regression model at 10 fold cross validation obtain high accurate predictive values of the angiogenic factors across different GA including validation.

However the demographic as well as the clinical characteristics may needed further inclusion of variables which could give meaningful interpretation of the result such as Race, BMI, IVF, smoking history singleton or multiparity and mentioning exclusion number of total patients during analysis and thereby upgrading the tables

Moreover the definition of Preeclampsia needs to be stated clearly ;

There are copple of sites where reference may needed such as lines 313,324,345

Finally the manuscript needs to be written more concisely- for example methods section containing study population or study design/preparation could be merged and shortened in one paragraph ; Also too much info on VEGF would not matters given that sflt1/plgf ratio itself has a good predictive value and gives similar result as other studies.

Reviewer #3: General Comments

The authors conducted a study assessing the predictive ability of angiogenic factors for pre-eclampsia. While this is an important area, this has been explored in several studies including in sub-Saharan countries (PMID: 28137987; PMID: 29523269); not referenced by the authors. Therefore, a clear justification of why this study was conducted and what it adds to the literature is needed, but not adequately provided. In general, the introduction is too long with a lot of basic information. Consider revising and focusing more on the focus of the study objective.

Specific comments:

Line 87- 89: The authors state that BP and urine measurements are not specific to pre-eclampsia, therefore warranting assessment of angiogenic factors. I don’t agree with this statement as angiogenic factors are not specific to pre-eclampsia only. Please revise or justify reasoning more.

Table 1: Add the total number of cases and controls to the top rows

The authors stated that they matched 106 each in cases and controls but some category numbers do not add up to this in the table e.g. gestational age.

Why bother grouping age into 43-47 when only person falls under this category?

The authors present adjusted ORs in Table 5 but do not state anywhere the variables that were adjusted.

6. PLOS authors have the option to publish the peer review history of their article (what does this mean?). If published, this will include your full peer review and any attached files.

Reviewer #1: **Yes: **Prof .Girija Wagh ,MD FICOG

Reviewer #2: **Yes: **saira salahuddin

Reviewer #3: **Yes: **Ugochinyere Vivian Ukah

---

## [Author Response · Author response to Decision Letter 0]

10 Feb 2021

Editor Comments:

Editor: Please ensure that your manuscript meets PLOS ONE's style requirements, including those for file Renaming. The PLOS ONE style templates can be found at

Response: We have reorganised all components of the manuscript to meet the PLOS One style requirements, including file renaming as guided by the templates at the provided links.

Editor: Please include additional information regarding the survey or questionnaire used in the study and ensure that you have provided sufficient details that others could replicate the analyses. For instance, if you developed a questionnaire as part of this study and it is not under a copyright more restrictive than CC-BY, please include a copy, in both the original language and English, as Supporting Information.

Response: We developed a data tool as part of the study. This was printed only in English and administered by a study nurse/midwife. We have included a copy of the data tool in the submission.

Editor: You indicated that you had ethical approval for your study. In your Methods section, please ensure you have also stated whether you obtained consent from parents or guardians of the minors included in the study or whether the research ethics committee or IRB specifically waived the need for their consent. 

Response: As per the Uganda National Council for Science and Technology, pregnant teenage mothers above 14 years of age are considered emancipated minors capable of giving informed consent (https://www.uncst.go.ug/guidelines-and-forms/National Guidelines for Research Involving Humans as Research Participants). We therefore had permission from the IRB to include participants below 18 years in the study, however none of these were included because they were ineligible for various other reasons. We have made the required corrections in the methods section.

Editor: Please provide a sample size and power calculation in the Methods, or discuss the reasons for not performing one before study initiation.

Response: We have included a section “Sample size estimation” in the methods section to indicate the sample size and power calculations.

Reviewers' comments:

Reviewer #1: 

Reviewer: The study has been conducted well to analyse the fact that these angio markers are raised in PE. The predictability is not assessed due to the fact that the inclusion criteria has been 20 weeks of gestation so the narration about this issue is out of context and needs to be modified.

Response: Thank you for this comment, we have revised/modified all the aspects of the manuscript to represent the correct context of this work, from the introduction up to the discussion.

Reviewer: Establishment of the fact that angiogenic markers are deranged in comparison to matched normal population is a good study. What you can do is look at the levels quantitatively and identify those one who have a count of 85 or more and compare them to find out whether they had a severe disease. this way rather than for prediction you can also understand its value in risk stratification which is clinically of great value. If you are interested you can request the gynecs in your institution to connect to me at girijawagh@gmail.com about a gestosis score validation may be can be done in-the same population through a different study with new ethics approval.

Response: Thank you for the suggestions for another study that can be carried out in this setting to investigate the role of angiogenic markers in risk stratification using a gestosis score. We are interested in pursuing this and we shall be getting in touch with you at the email provided for further discussion. 

Reviewer #2: 

Reviewer: This is a case control study that reflects the association of maternal circulating levels of VEGF, PlGF, sFlt1 and the sFlt1/PlGF ratio with PE, and compare their distribution both in normal healthy pregnancies as well as those with PE and to determine their predictive power of the disease among women in a Ugandan population. The uniqueness of the study is employing different methodological approaches such as Luminex assay, heat map, trained logistical regression model at 10 fold cross validation obtain high accurate predictive values of the angiogenic factors across different GA including validation. However the demographic as well as the clinical characteristics may needed further inclusion of variables which could give meaningful interpretation of the result such as Race, BMI, IVF, smoking history singleton or multiparity and mentioning exclusion number of total patients during analysis and thereby upgrading the tables -

Response: We have included family history of diabetes mellitus and smoking history to the clinical characteristics in Table 2. All participants self-reported to be Black Africans with both parents of African ancestry. Further ancestral delineation was not feasible. All participants had conceived naturally. We have adjusted the tables to include the total number of participants. 

Reviewer: Moreover the definition of Preeclampsia needs to be stated clearly

Response: We have made corrections to the definition of Preeclampsia, stating; Preeclampsia was defined as new onset hypertension consisting of increased systolic BP of ≥ 140 mmHg and diastolic BP ≥ 90 mmHg on 2 different measurements at least 4 hours apart, plus proteinuria ≥ +1 on dipstick, at ≥ 20 weeks of gestation. 

Reviewer: There are copple of sites where reference may needed such as lines 313,324,345

Response: We have made changes to reduce the discussion on VEGF as advised by another reviewer. Hence we deleted the above sentences requiring referencing. We have appropriately referenced the rest of the discussion. 

Reviewer: Finally the manuscript needs to be written more concisely- for example methods section containing study population or study design/preparation could be merged and shortened in one paragraph; Also too much info on VEGF would not matters given that sFlt1/PlGF ratio itself has a good predictive value and gives similar result as other studies.

Response: We have made corrections and shortened the section on study design and study population. 

Reviewer#3: 

General Comments

Reviewer: The authors conducted a study assessing the predictive ability of angiogenic factors for pre-eclampsia. While this is an important area, this has been explored in several studies including in sub-Saharan countries (PMID: 28137987; PMID: 29523269); not referenced by the authors. Therefore, a clear justification of why this study was conducted and what it adds to the literature is needed, but not adequately provided.

Response: We have provided a better justification of the study and have also included the suggested references.

Reviewer: In general, the introduction is too long with a lot of basic information. Consider revising and focusing more on the focus of the study objective.

Response: Thank you for the comment, the introduction has been reduced and made more focused.

Reviewer: Line 87- 89: The authors state that BP and urine measurements are not specific to pre-eclampsia, therefore warranting assessment of angiogenic factors. I don’t agree with this statement as angiogenic factors are not specific to pre-eclampsia only. Please revise or justify reasoning more

Response: We have revised and also deleted that statement.

Reviewer: Table 1: Add the total number of cases and controls to the top rows

Response: We have included the total numbers of cases and controls.

Reviewer: The authors stated that they matched 106 each in cases and controls but some category numbers do not add up to this in the table e.g., gestational age.

Response: We have checked and updated Table 1 accordingly.

Reviewer: Why bother grouping age into 43-47 when only person falls under this category? 

Response: We have changed age group 33-37 to Above 32 years. As such, the person previously grouped under 43-47 has been included in the Above 32 years category.

Reviewer The authors present adjusted ORs in Table 5 but do not state anywhere the variables that were adjusted. 

Response: We have added a statement to reflect under the section; ‘Association of angiogenic factors with Preeclampsia’, “In the multivariate analysis, we adjusted for family history of preeclampsia and family history of hypertension”. This analysis has been included in Table 5.

---

## [Decision Letter · Decision Letter 1]

8 Apr 2021

PONE-D-20-27867R1

Circulating levels of angiogenic factors and their association with preeclampsia among pregnant women at Mulago national referral hospital in Uganda

PLOS ONE

Dear Dr. Nakimuli,

Thank you for submitting your manuscript to PLOS ONE. After careful consideration, we feel that it has merit but does not fully meet PLOS ONE’s publication criteria as it currently stands. Therefore, we invite you to submit a revised version of the manuscript that addresses the points raised during the review process.

We look forward to receiving your revised manuscript.

Kind regards,

Antonio Simone Laganà, M.D., Ph.D.

Academic Editor

PLOS ONE

Journal Requirements:

Additional Editor Comments (if provided):

I appreciate the efforts of the authors in making the changes, as recommended by the reviewers.

Nevertheless, two reviewers still have some concerns: for this reason, I invite the authors to perform these minor changes and resubmit the manuscript.

Reviewers' comments:

Reviewer's Responses to Questions

**Comments to the Author**

1. If the authors have adequately addressed your comments raised in a previous round of review and you feel that this manuscript is now acceptable for publication, you may indicate that here to bypass the “Comments to the Author” section, enter your conflict of interest statement in the “Confidential to Editor” section, and submit your "Accept" recommendation.

Reviewer #1: All comments have been addressed

Reviewer #2: (No Response)

Reviewer #3: (No Response)

2. Is the manuscript technically sound, and do the data support the conclusions?

Reviewer #1: Partly

Reviewer #2: (No Response)

Reviewer #3: Yes

3. Has the statistical analysis been performed appropriately and rigorously? 

Reviewer #1: Yes

Reviewer #2: (No Response)

Reviewer #3: No

4. Have the authors made all data underlying the findings in their manuscript fully available?

Reviewer #1: Yes

Reviewer #2: (No Response)

Reviewer #3: Yes

5. Is the manuscript presented in an intelligible fashion and written in standard English?

Reviewer #1: Yes

Reviewer #2: (No Response)

Reviewer #3: Yes

6. Review Comments to the Author

Reviewer #1: THE SENTENCES NUMBERED FORM 205 TO 213 NEED CLARITY AS THEY ARE VITAL TO UNDERSTAND .PLEASE WRITE THE CORRECT INFORMATION AS THE SENTENCE FOR BOTH CASES AND CONTROLS ARE THE SAME

TOO MUCH TECHNICALITY ABOUT THE ASSAYS CAN EB CONSOLIDATED

ELABORATION ABOUT TRANSLATION IN CLINICAL PRACTIE IS IMPORTANT TO MAKE THIS STUDY USEFUL

Reviewer #2: (No Response)

Reviewer #3: Table 5 - Why did the authors not include other variables in the model adjustment such as maternal age at pregnancy? Also add a footnote with the adjusted variables under the table 5.

7. PLOS authors have the option to publish the peer review history of their article (what does this mean?). If published, this will include your full peer review and any attached files.

Reviewer #1: **Yes: **Girija Wagh

Reviewer #2: **Yes: **saira salahuddin

Reviewer #3: No

---

## [Author Response · Author response to Decision Letter 1]

20 Apr 2021

Editor’s comments: 

Editor: Journal Requirements: Please review your reference list to ensure that it is complete and correct. If you have cited papers that have been retracted, please include the rationale for doing so in the manuscript text, or remove these references and replace them with relevant current references. Any changes to the reference list should be mentioned in the rebuttal letter that accompanies your revised manuscript. If you need to cite a retracted article, indicate the article’s retracted status in the References list and also include a citation and full reference for the retraction notice.

Response: We have reviewed the reference list and affirm that it is complete and correct. There are no retracted papers cited. 

Reviewers’ comments: 

Reviewer #1: Reviewer: THE SENTENCES NUMBERED FROM 205 TO 213 NEED CLARITY AS THEY ARE VITAL TO UNDERSTAND. PLEASE WRITE THE CORRECT INFORMATION AS THE SENTENCE FOR BOTH CASES AND CONTROLS ARE THE SAME. 

Response: Thank you for this comment. We have corrected the sentences indicated and now read more clearly. 

Reviewer: TOO MUCH TECHNICALITY ABOUT THE ASSAYS CAN EB CONSOLIDATED. 

Response: We have summarised all the laboratory procedures under one paragraph “Plasma collection and immunoassays”. 

Reviewer: ELABORATION ABOUT TRANSLATION IN CLINICAL PRACTIE IS IMPORTANT TO MAKE THIS STUDY USEFUL. 

Response: Thank you for this comment, we have addressed this in the discussion, lines 388 to 392 in the manuscript with track changes, to elaborate how the results obtained can be translated into clinical practice in Uganda. 

Reviewer #3: 

Reviewer: Table 5 - Why did the authors not include other variables in the model adjustment such as maternal age at pregnancy? 

Response: Thank you so much for this comment. We started with a univariate analysis of all the variables against the dependent variable. Variables with a p-value less than 0.25 in univariate analysis were considered for the multivariate analysis. The variables used in the final model were obtained by backward elimination. We have added this to the main text, Lines 273 to 275 in the manuscript with track changes. Furthermore, maternal age could not be included in the model because it was one of the variables used to match cases to their controls. 

Reviewer: Also add a footnote with the adjusted variables under the table 5. 

Response: Thank you so much for this comment. Adjusted variables have been indicated under Table 5 as advised.

---

## [Editor Report · Decision Letter 2]

23 Apr 2021

Circulating levels of angiogenic factors and their association with preeclampsia among pregnant women at Mulago national referral hospital in Uganda

PONE-D-20-27867R2

Dear Dr. Nakimuli,

We’re pleased to inform you that your manuscript has been judged scientifically suitable for publication and will be formally accepted for publication once it meets all outstanding technical requirements.

Kind regards,

Antonio Simone Laganà, M.D., Ph.D.

Academic Editor

PLOS ONE

Additional Editor Comments (optional):

Authors performed the required corrections. I am pleased to accept this paper for publication.
---

## [Editor Report · Acceptance letter]

29 Apr 2021

PONE-D-20-27867R2 

Circulating levels of angiogenic factors and their association with preeclampsia among pregnant women at Mulago national referral hospital in Uganda 

Dear Dr. Nakimuli:

I'm pleased to inform you that your manuscript has been deemed suitable for publication in PLOS ONE. Congratulations! Your manuscript is now with our production department. 

Kind regards, 

on behalf of

Dr. Antonio Simone Laganà 

Academic Editor

PLOS ONE